# Space–Time Clustering Characteristics of Malaria in Bhutan at the End Stages of Elimination

**DOI:** 10.3390/ijerph18115553

**Published:** 2021-05-22

**Authors:** Kinley Wangdi, Kinley Penjor, Saranath Lawpoolsri, Ric N. Price, Peter W. Gething, Darren J. Gray, Elivelton Da Silva Fonseca, Archie C. A. Clements

**Affiliations:** 1Research School of Population Health, Australian National University, Canberra, ACT 2601, Australia; darren.gray@anu.edu.au; 2Vector-Borne Diseases Control Program, Department of Public Health, Ministry of Health, Gelephu 31101, Bhutan; kinleyp@health.gov.bt (K.P.); tobgye@health.gov.bt (T.); 3Department of Tropical Hygiene, Faculty of Tropical Medicine, Mahidol University, Bangkok 10400, Thailand; saranath.law@mahidol.ac.th; 4Global and Tropical Health Division, Menzies School of Health Research, Charles Darwin University, Darwin, NT 0810, Australia; ric.price@menzies.edu.au; 5Centre for Tropical Medicine and Global Health, Nuffield Department of Clinical Medicine, University of Oxford, Oxford OX1 2JD, UK; 6Mahidol-Oxford Tropical Medicine Research Unit, Faculty of Tropical Medicine, Mahidol University, Bangkok 10400, Thailand; 7Faculty of Health Sciences, Curtin University, Bentley, WA 6102, Australia; peter.gething@telethonkids.org.au (P.W.G.); archie.clements@curtin.edu.au (A.C.A.C.); 8Telethon Kids Institute, Nedlands, WA 6009, Australia; 9Institute of Geography, Federal University of Uberlândia, Uberlândia, MG 38408-100, Brazil; elivelton.fonseca@gmail.com

**Keywords:** Bhutan, malaria, space, time, clustering, SaTScan

## Abstract

Malaria in Bhutan has fallen significantly over the last decade. As Bhutan attempts to eliminate malaria in 2022, this study aimed to characterize the space–time clustering of malaria from 2010 to 2019. Malaria data were obtained from the Bhutan Vector-Borne Disease Control Program data repository. Spatial and space–time cluster analyses of *Plasmodium falciparum* and *Plasmodium vivax* cases were conducted at the sub-district level from 2010 to 2019 using Kulldorff’s space–time scan statistic. A total of 768 confirmed malaria cases, including 454 (59%) *P. vivax* cases, were reported in Bhutan during the study period. Significant temporal clusters of cases caused by both species were identified between April and September. The most likely spatial clusters were detected in the central part of Bhutan throughout the study period. The most likely space–time cluster was in Sarpang District and neighboring districts between January 2010 to June 2012 for cases of infection with both species. The most likely cluster for *P. falciparum* infection had a radius of 50.4 km and included 26 sub-districts with a relative risk (RR) of 32.7. The most likely cluster for *P. vivax* infection had a radius of 33.6 km with 11 sub-districts and RR of 27.7. Three secondary space–time clusters were detected in other parts of Bhutan. Spatial and space–time cluster analysis identified high-risk areas and periods for both *P. vivax* and *P. falciparum* malaria. Both malaria types showed significant spatial and spatiotemporal variations. Operational research to understand the drivers of residual transmission in hotspot sub-districts will help to overcome the final challenges of malaria elimination in Bhutan.

## 1. Introduction

Historically, malaria in Bhutan has been reported from seven out of twenty districts: Chukha, Dagana, Pemagatshel, Samdrup Jongkhar, Samtse, Sarpang and Zhemgang [1]. These districts lie in the southern foothills of the Himalayas bordering the Indian states of Assam and West Bengal. The main malaria parasite species are *Plasmodium falciparum* and *Plasmodium vivax*, and the main suspected vector species are *Anopheles pseudowillmori* and *Anopheles culicifacies* [2]. Malaria cases have been falling in recent years. As a result, Bhutan announced a national strategy to eliminate malaria by 2022 [3,4]. The success of malaria control in Bhutan can be attributed to the implementation of a three-pronged approach, namely: (1) universal access to early diagnosis and prompt treatment with artemisinin-based combination therapy (ACT); (2) the protection of at-risk populations with indoor residual spraying (IRS) and long-lasting insecticide nets (LLINs), and (3) integrated vector management (IVM).

As the malaria map shrinks, cases tend to be confined to hard-to-reach populations, often in border areas [5,6]. Identifying these last pockets of transmission is imperative for the program to eliminate the parasite and prevent its reintroduction. Spatial epidemiological tools (including geographical information systems (GIS) and spatial analytic methods) can be used to quantify spatial and spatiotemporal patterns of malaria, including clustering [7,8,9,10]. The identification of malaria clusters can help in the delineation of problem areas and the application of targeted program interventions suitable in the elimination phase [11]. Focused interventions in areas with a higher risk of malaria are likely to be more cost-effective than uniform resource allocation, particularly in resource-constrained settings for sustainable elimination programs [9,12,13]. In addition, knowledge of seasonal patterns will aid in the estimation of time for malaria transmission for initiating suitable and appropriate control measures [14,15].

The aims of this study were to quantify the temporal, spatial and spatial–temporal patterns of both species of malaria in sub-districts in Bhutan using retrospective surveillance data.

## 2. Methods

### 2.1. Study Site

Bhutan is a small Himalayan country in South Asia, neighboring India in the east, south and west, and China in the north. The country is divided into 20 districts and 205 sub-districts. The total population of Bhutan in 2017 was estimated at 681,720 [16]. There is a wide variation in the climate, with sub-tropical conditions in the southern foothills neighboring India to snow-capped mountains rising above 4500 m from sea level in the north. Malaria transmission occurs mainly in southern Bhutan in 82 sub-districts within the seven historically malaria-endemic districts (Figure 1).

### 2.2. Data Source

Malaria data for the study were obtained from the national malaria surveillance system, housed within the Bhutan Vector-borne Diseases Control Program (VDCP). Monthly malaria cases for 2010–2019 from 82 sub-districts were extracted, including those with zero cases. In these malaria-transmitting sub-districts, all febrile patients presenting to health facilities (hospitals and primary health centers) undergo a mandatory blood test with RDTs or microscopy. Malaria cases stratified by species as *P. falciparum* and *P. vivax* are reported through the national malaria surveillance system at the end of each month. All patients in public health facilities are offered treatment free of charge according to national guidelines. The yearly sub-district population was calculated using the district population growth rate (Chukha 1.7, Dagana 1.9, Pemagatshel 1.2, Samdrup Jongkhar 1.8, Samtse 1.6, Sarpang 2.0 and Zhemgang 1.4) as outlined by the National Statistical Bureau and the Office of the Census Commissioner of Bhutan [17,18]. An electronic map of sub-district boundaries in shapefile format was obtained from the Global Administrative Areas database (http://www.gadm.org/country, accessed on 21 December 2020).

### 2.3. Exploration of Seasonal Patterns and Inter-Annual Patterns

The malaria time series (January 2010–December 2019) was decomposed using seasonal trend decomposition based on locally (STL) weighted regression. The STL model was structured as follows:(1)$$Y_{t}=S_{t}+T_{t}+R_{t}$$
where *Y_t_* represents numbers of local malaria cases with logarithmic transformation, *S_t_* is the additive seasonal component, *T_t_* is the trend, and *R_t_* is the “remainder component”; *t* is time in months [19,20,21].

### 2.4. Spatial and Spatiotemporal Cluster Analysis

Kulldorff’s spatial and space–time scan statistics were applied to identify malaria incidence clusters by species (*P. falciparum* and *P. vivax*) at a sub-district level over the 10-year study period. The analysis was implemented using the SaTScan^TM^ software v.9.4.2 (M Kulldorf and Information Management Services Inc., Calverton, MD, USA) [22]. Three data sets were inputted to the program for analysis: the shapefile centroids of the sub-districts, using the latitude and longitude in the World Geodetic Coordinate System of 1984, yearly populations and reported cases of 82 sub-districts.

Spatial cluster analysis for each year were identified using a Poisson variant of the spatial scan statistic. The following configurations were used in the analysis: clusters of maximum size equal to 25% of the exposed population, no geographical overlapping of clusters and 999 Monte Carlo replications for the testing of statistical significance to ensure adequate power for defining clusters [23,24]. In the space–time cluster analysis, the unit of time was a month, with same configurations as the spatial analysis. SaTScan uses moving scanning windows (circular or elliptical) of varying sizes to estimate the probability that the frequency of positive cases within a window is greater than what is expected by chance. It takes into account the observed number of cases inside and outside the windows, to detect clusters and estimate relative risk (RR). For each location, the window size with the highest log-likelihood ratio (LLR) was considered the most probable cluster, i.e., the cluster that is least likely to have occurred randomly. The SaTScan output for statistically significant clusters includes the location of the center of the scanning window, the radius of the scanning window, the number of observed and expected positives within the circle and the relative risk, LLR and *p* value of the cluster. Statistically significant clusters were defined as those with a *p* < 0.05. Centroids of the districts and significant clusters were mapped using the GIS ArcMap 10.5.1 (ESRI, Redlands, CA, USA).

## 3. Results

### 3.1. Characteristics of the Study Population

Between 2010 and 2019, a total of 768 confirmed malaria cases were reported, including 314 (41%) patients infected with *P. falciparum* and 454 (59%) with *P. vivax*. Forty-three percent (330) were in the age group of 21–45 years and only 1.3% (10) were children under one year of age. Males made up 65.1% (500) of patients and farmers and students constituted 47.3% (363) and 25.0% (192), respectively. The years with the highest and lowest numbers of cases were 2010 and 2013, with 52.2% (401) and 1.6% (12), respectively. Sarpang, Samdrup Jongkhar and Samtse reported 68.1% (523), 10.3% (79) and 6.5% (65) of all cases (Table 1). The yearly annual parasite incidence (API) of sub-districts (cases of sub-district divided by population per 1000) is shown in Appendix A

### 3.2. Time-Series Decompositions

The time-series decompositions show a clear seasonal pattern which is evident in the raw data for both *P. falciparum* and *P. vivax*. A large peak occurred for *P. falciparum* in May, followed by two smaller peaks later in the year. For *P. vivax,* a large peak occurred in May, followed by a plateauing for the subsequent three months and dropping off during the remainder of the rainy season. The inter-annual pattern showed a large peak in 2010, with a lower incidence in subsequent years for both species of malaria (Figure 2).

### 3.3. Purely Spatial Analysis

The most likely clusters of *P. falciparum* infections were in the central part of southern Bhutan for eight of the ten years of the study period (Table 2). In 2014 and 2019, the most likely clusters shifted to the eastern part of Bhutan, in the sub-district of Langchenphug. There were three secondary clusters in 2010, 2012 and 2014, respectively (Figure 3). No cluster was observed in 2016 or 2017.

Similar to *P. falciparum*, the most likely clusters of *P. vivax* infection were in the central part of Bhutan, except in 2017, when it was in the western part of Bhutan (Table 3). There was a secondary cluster in 2010 in the eastern part of Bhutan (Figure 4). The largest radius cluster was observed in 2010 for both species.

### 3.4. Spatiotemporal Analysis

The most likely spatiotemporal cluster was in the central part of the country between January 2010 to June 2012 for both species (Figure 5). The most likely cluster of *P. falciparum* infection had a radius of 50.4 km and included 26 sub-districts with an RR of 32.7. Two secondary clusters in April 2014 had a radius of 0 (indicating a single sub-district; RR = 25.4) and 56.8 km (RR = 16.7), respectively. Another secondary cluster was detected from January 2010 to December 2014 in the far east of Bhutan (one district; RR 25.4).

The location and period of the most likely cluster for *P. vivax* was similar to *P. falciparum* but it had a smaller radius of 33.6 km, contained 11 sub-districts and had an RR of 27.7. Two secondary clusters were observed from March–May 2010 with a radius of 9.5 km (two sub-districts; RR = 104.7) and 13.1 km (seven sub-districts; RR = 17.9). The third secondary cluster was observed in August–November 2017 with one sub-district and an RR of 21.3 (Table 4 and Table 5).

## 4. Discussion

This 10-year retrospective analysis of malaria surveillance data from Bhutan provides an in-depth characterization of significant spatial and spatiotemporal clustering of both *P. falciparum* and *P. vivax* malaria incidence. Both species showed a clear seasonal pattern with a large peak in the early part of the year. Malaria clusters of both species were observed in the central parts of Bhutan along the international border throughout the study period. The most likely spatiotemporal clusters were identified from January 2010–June 2012 for both species of malaria in Sarpang District. There were three secondary clusters for both species of malaria in other districts.

The seasonal pattern corresponds with the monsoon in Bhutan, which lasts for five months from May to September [25]. The monsoon season supports the breeding and growth of the main malaria vectors in Bhutan; *Anopheles pseudowillmori* and *Anopheles Culicifacies* [2,26]. Therefore, VDCP can utilize this knowledge to implement focused interventions of malaria including the distribution of LLINs and IRS. Even though LLIN use was reported to be high in Bhutan [27], monitoring the use of LLINs should be strengthened, especially during these months, to interrupt local transmission.

The spatial cluster analysis identified high-risk sub-districts along the international border with India. As Bhutan approaches malaria elimination, a key challenge will be residual malaria in the population living in the border area, highlighting the need for novel approaches to address the problem. Foreign nationals entering Bhutan through the land crossings in the southern districts include two groups: those staying in Bhutan on a long-term basis (such as Indians working on developmental projects) and those visiting Bhutan during day visits for business and employment [2]. Between 2006–2014, daily visitors crossing the southern border with India for employment accounted for 13% of all malaria cases [2]. In light of ongoing transmission in Sarpang District, the VDCP will need to monitor the incidence of malaria in this migrant population closely, using both passive and active case detection to better understand local transmission dynamics. Further, active blood surveys in the hotspot sub-districts, followed up by targeted interventions, should be started in earnest to disrupt local transmission.

The lack of spatial clusters of *P. falciparum* in 2016 and 2017 could be attributed to the effects of LLINs which are distributed every three years, and which might have disrupted the natural transmission dynamics of malaria in that year. Since the start of the LLIN distribution program in 2006, five rounds of LLINs were completed in 2006, 2010, 2014, 2017 and 2020. In 2014, LLIN coverage was 99.0%, and 94% of people slept regularly under LLINs [27].

The space–time analysis identified a primary cluster in Sarpang District from January 2010–June 2012. Sarpang continues to be the location of the remaining hotspot of malaria in Bhutan [1,25]. The elimination of malaria from Sarpang has proven more difficult as compared to other districts. A major reason is likely to be the proximity of villages in Sarpang to the very long and porous border with the Indian states of Assam [2,28]. The risk of transmission of malaria across the border within the flight range of infected mosquitoes in these villages is high. Control measures across the international border (in Assam) are sporadically implemented and transmission is maintained due to the local populations living in villages near to forest fringe areas [29]. Cross-border movement has been acknowledged as one of the key challenges in malaria elimination [5,13,28], leading to a series of cross-border meetings being held to strengthen collaboration between India and Bhutan to achieve malaria elimination [30]. However, the presence of a *P. falciparum* cluster in Langchenphu (in Samdrup Jongkhar District) should precipitate a call for the VDCP to invest time and resources to study the local drivers of malaria.

A secondary space–time cluster of *P. vivax* from August to November 2017 was observed in Samtse District. This was due to an outbreak of *P. vivax* in Samtse with 16 cases reported over three months. A WHO external review of malaria surveillance reported that the outbreak was due to the late diagnosis of indigenous cases with a consequent delay in response. As malaria cases dwindle in Bhutan, people are less likely to seek care for malaria as well as use protective measures regularly. Therefore, the prompt diagnosis of malaria and adherence to the regular use of LLINs should be reinforced through regular health education. Further, primaquine is used for the radical cure of *P. vivax* in Bhutan [31], although no studies on adherence to the recommended 14-day radical cure regimen have been undertaken. This will be an important step as the country attempts to eliminate malaria by 2022. Poor adherence has been reported to be an important factor that drives *P. vivax* transmission in other parts of the world [32]. It is timely for the VDCP to undertake operational research on the barriers and enablers of the current radical cure regimen, including directly observed treatment (DOT) recommended for *P. vivax* in Bhutan, to develop strategies to improve adherence.

Our study has a number of important limitations. The main limitations are inherent to the method used for the space–time analysis, especially the scale (sub-district) at which data were available. Even though SaTScan is a powerful statistical tool frequently used to analyze spatial and spatio-temporal patterns of disease, it is a challenge to determine an optimal set of scaling parameters for the SaTScan analysis, especially for the maximum window size. Indeed, large maximum window sizes (such as 50% of the total population) can hide small, homogeneous clusters within the larger and more heterogeneous ones, while on the other hand, small maximum window sizes can miss significant regional-level clusters. Second, the completeness and representativeness of surveillance data cannot be ascertained. However, this is likely to be minimal since all malaria cases are treated only in public health facilities in Bhutan (there is no private health care sector) and they report routinely to the national malaria surveillance system. Third, sub-district populations were not available, and they were projected based on the National Housing and Population Census 2007. However, the main strength of our study is the breakdown of the monthly malaria incidence, stratified by species at a fine-resolution (sub-districts), providing a more precise space–time analysis.

In conclusion, malaria has been eliminated in most sub-districts in Bhutan and a few remaining clusters were largely confined to the southern district of Sarpang. The VDCP of Bhutan continues to plan for the sub-national elimination of malaria while awaiting national elimination. Operational research to understand the burden of malaria in daily migrant workers and transmission dynamics in hotspot sub-districts could be valuable to overcome the last mile challenge of malaria elimination in Bhutan.

## Figures and Tables

**Figure 1 ijerph-18-05553-f001:**
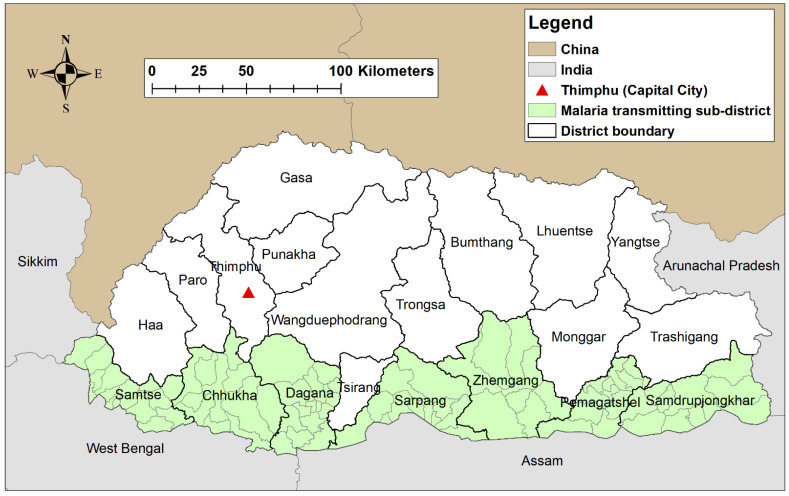
Map of Bhutan with malaria-transmitting districts and sub-districts.

**Figure 2 ijerph-18-05553-f002:**
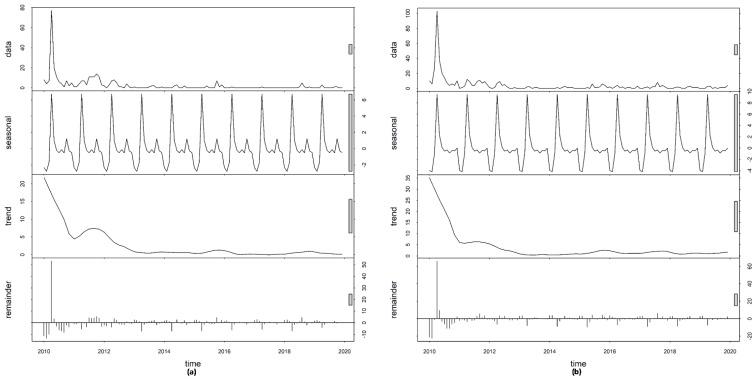
Decomposed (**a**) *Plasmodium falciparum* and (**b**) *Plasmodium vivax* cases of Bhutan, 2010–2019.

**Figure 3 ijerph-18-05553-f003:**
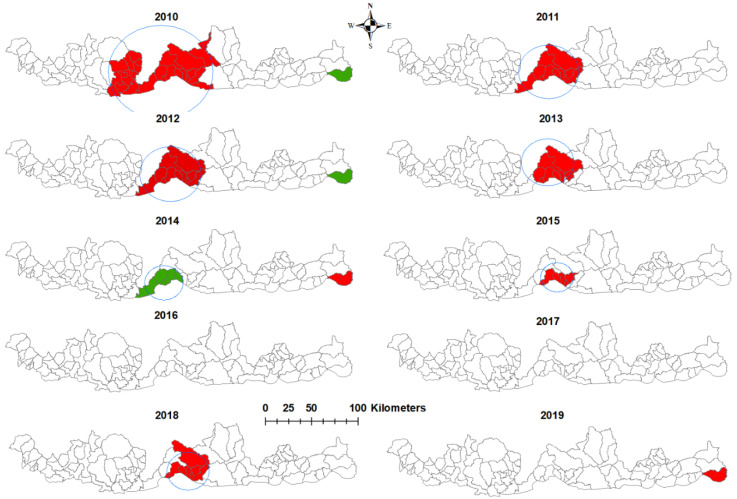
Spatial clusters of *Plasmodium falciparum* malaria in Bhutan by years, 2010–2019. (Red- most likely cluster; Green- secondary cluster).

**Figure 4 ijerph-18-05553-f004:**
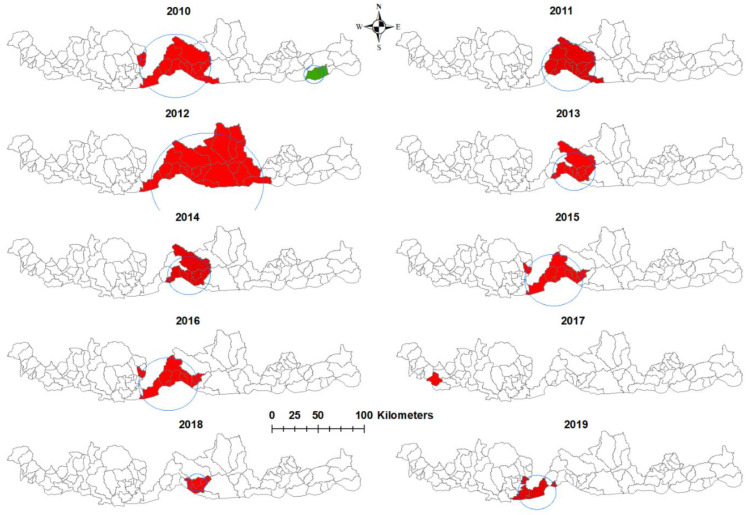
Spatial clusters of *Plasmodium vivax* malaria in Bhutan by years, 2010–2019. (Red- most likely cluster; Green- secondary cluster).

**Figure 5 ijerph-18-05553-f005:**
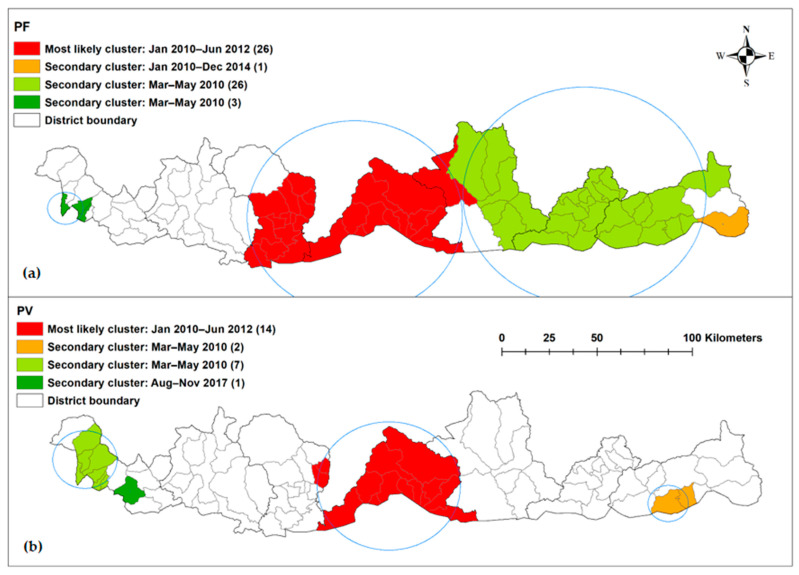
Spatiotemporal clusters of (**a**) *Plasmodium falciparum* and (**b**) *Plasmodium vivax*, in Bhutan 2010–2019.

**Table 1 ijerph-18-05553-t001:** Socio-demographic characteristics of *Plasmodium falciparum* and *Plasmodium vivax* during the study period.

Characteristics	*Plasmodium falciparum* (%)	*Plasmodium vivax* (%)	Total (%)
**Age (years)**			
≤1	5 (1.6)	5 (1.1)	10 (1.3)
2–12	63 (20.1)	75 (16.5)	138 (18.0)
13–20	61 (19.4)	75 (16.5)	136 (17.7)
21–45	127 (40.5)	203 (44.7)	330 (43.0)
>45	58 (18.5)	96 (21.2)	154 (20.1)
**Sex**			
Female	107 (34.1)	161 (35.5)	268 (34.9)
Male	207 (65.9)	293 (64.5)	500 (65.1)
**Occupation**			
Business	4 (1.3)	9 (2.0)	13 (1.7)
Dependent	23 (7.3)	23 (5.1)	46 (6.0)
Farmer	146 (46.5)	217 (47.8)	363 (47.3)
House wife	24 (7.6)	40 (8.8)	64 (8.3)
Labour	5 (1.6)	11 (2.4)	16 (2.1)
Monk	2 (0.6)	4 (0.9)	6 (0.8)
Public servant	7 (2.2)	26 (8.1)	33 (4.3)
Armed forces	17 (5.4)	18 (2.4)	35 (4.6)
Student	86 (27.4)	106 (23.4)	192 (25.0)
**Year**			
2010	152 (48.4)	249 (54.9)	401 (52.2)
2011	84 (26.8)	74 (16.3)	158 (20.6)
2012	32 (10.2)	30 (6.6)	62 (8.1)
2013	6 (1.9)	6 (1.3)	12 (1.6)
2014	11 (3.5)	8 (1.8)	19 (2.5)
2015	13 (4.1)	22 (4.9)	35 (4.6)
2016	2 (0.6)	17 (3.7)	19 (2.5)
2017	1 (0.3)	23 (5.1)	24 (3.1)
2018	8 (2.6)	12 (2.6)	20 (2.6)
2019	5 (1.6)	13 (2.9)	18 (2.3)
**District**			
Chukha	5 (1.6)	22 (4.9)	27 (3.5)
Dagana	26 (8.3)	17 (3.7)	43 (5.6)
Pemagatshel	6 (1.9)	13 (2.9)	19 (2.5)
Samdrup Jongkhar	43 (13.7)	36 (7.9)	79 (10.3)
Samtse	13 (4.1)	52 (11.5)	65 (8.5)
Sarpang	217 (69.1)	306 (67.4)	523 (68.1)
Zhemgang	4 (1.3)	8 (1.8)	12 (1.6)

**Table 2 ijerph-18-05553-t002:** Spatial clusters of *Plasmodium falciparum* in Bhutan, 2010–2019.

Year	SaTScan Statistics	Long	Lat	Radius (km)	Population	Number of Cases	Expected Cases	No of Sub-Districts	RR	LLR	*p*-Value
2010	Most likely cluster	90.2587	26.9402	50.36	70,171	120	36.51	26	11.86	101.71	<0.001
	Secondary cluster	91.9991	26.9079	0	2067	13	1.08	1	13.12	20.96	<0.001
2011	Most likely cluster	90.3536	26.9405	29.51	41,701	60	11.79	11	15.31	71.18	<0.001
2012	Most likely cluster	90.3536	26.9405	29.51	42,535	26	4.51	11	26.44	36.44	<0.001
	Secondary cluster	91.9991	26.9079	0	2143	4	0.23	1	19.99	7.94	<0.001
2013	Most likely cluster	90.3435	27.0545	25.79	37,861	6	0.74	8	Infinity	12.56	<0.001
2014	Most likely cluster	91.9991	26.9079	0	2220	6	0.078	1	167.64	22.14	<0.001
	Secondary cluster	90.3536	26.9405	9.4	14,771	5	0.52	3	16.79	7.97	<0.016
2015	Most likely cluster	90.4328	26.9283	15.71	32,603	11	1.33	6	48.10	19.68	<0.001
2016	No cluster										
2017	No cluster										
2018	Most likely cluster	90.4845	26.9197	20.26	410,140	7	0.98	8	50.01	11.80	<0.001
2019	Most likely cluster	90.5214	26.8637	0	2428	3	0.036	1	208.35	11.47	<0.001

Lat—latitude; long—longitude; RR—relative risk; LLR—log-likelihood ratio.

**Table 3 ijerph-18-05553-t003:** Spatial clusters of *Plasmodium vivax* at sub-district level in Bhutan, 2006–2019.

Year	SaTScan Statistics	Long	Lat	Radius (km)	Population	Number of Cases	Expected Cases	No of Sub-Districts	RR	LLR	*p*-Value
2010	Most likely cluster	90.3536	26.9405	33.61	45,511	1166	38.79	14	10.84	164.19	<0.001
	Secondary cluster	91.6777	26.8583	9.49	4652	19	3.97	2	5.10	15.21	<0.001
2011	Most likely cluster	90.4328	26.9283	26.02	40,946	45	10.2	11	9.71	43.93	<0.001
2012	Most likely cluster	90.6599	26.8108	53.39	65,886	28	6.54	26	50.19	35.78	<0.001
2013	Most likely cluster	90.4845	26.9197	20.26	37,147	6	0.73	8	Infinity	12.78	<0.001
2014	Most likely cluster	90.4845	26.99197	20.26	37,913	7	0.97	8	50.68	11.88	<0.001
2015	Most likely cluster	90.2869	26.8729	28.05	42,318	20	2.93	10	65.08	33.90	<0.001
2016	Most likely cluster	90.2869	26.8729	28.05	43,161	14	2.27	10	30.26	20.69	<0.001
2017	Most likely cluster	89.1271	26.9157	0	9974	10	0.70	1	24.56	19.60	<0.001
2018	Most likely cluster	90.5628	26.9293	8.99	24,825	8	0.89	4	24.91	13.47	<0.001
2019	Most likely cluster	90.1291	26.7959	18.46	8678	7	0.33	4	44.50	16.85	<0.001

Lat—latitude; long—longitude; RR—relative risk; LLR—log-likelihood ratio.

**Table 4 ijerph-18-05553-t004:** Significant *Plasmodium falciparum* space–time clusters at sub-district level in Bhutan, 2010–2019.

Time Period (Month Year)	SaTScan Statistics	Long	Lat	Radius (km)	Population	Number of Cases	Expected Cases	No of Sub-Districts	RR	LLR	*p*-Value
January 2010–June 2012	Most likely cluster	90.2587	26.9402	50.36	76,572	207	17.71	26	32.66	399.91	<0.001
January 2010–December 2014	Secondary cluster	91.9991	26.9079	0	2243	25	1.07	1	25.37	55.86	<0.001
April 2010	Secondary cluster	91.3444	27.0405	56.81	72,597	9	0.55	26	16.69	16.76	<0.001
April 2010	Secondary cluster	88.8911	26.9823	8.33	11,685	5	0.089	3	57.15	15.28	0.0057

Lat—latitude; long—longitude; RR—relative risk; LLR—log-likelihood ratio.

**Table 5 ijerph-18-05553-t005:** Significant *Plasmodium vivax* space–time clusters at sub-district level in Bhutan, 2010–2019.

Time Period (Month Year)	SaTScan Statistics	Long	Lat	Radius (km)	Population	Number of Cases	Expected Cases	No of Sub-Districts	RR	LLR	*p*-Value
January 2010–June 2012	Most likely cluster	90.3536	26.9405	33.61	49,858	233	16.62	11	27.74	464.34	<0.001
March–May 2010	Secondary cluster	91.6777	26.8583	9.49	5048	17	0.17	2	104.65	61.91	<0.001
March–May 2010	Secondary cluster	88.9128	27.0651	13.09	25,715	15	0.87	7	17.85	28.85	<0.001
August–November 2017	Secondary cluster	89.1271	26.9157	0	9596	10	0.48	1	21.30	20.95	<0.001

Lat—latitude; long—longitude; RR—relative risk; LLR—log-likelihood ratio.

## Data Availability

Data supporting the results can be obtained upon request from Ministry of Health, Bhutan.

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
