# Peer review of "Space–Time Clustering Characteristics of Malaria in Bhutan at the End Stages of Elimination"

_ijerph, 2021, doi:10.3390/ijerph18115553_

Round 1
Reviewer 1 Report
See my attachment

Author Response
Reviewer #1
- The objective this study was “to characterise spatial and spatial-temporal clustering of cases caused by both species of malaria at sub-district level in Bhutan, to inform and guide malaria interventions as part of achieving malaria elimination goals.”
My comments: I will advise a revision of the above objective because this study has been conducted at a time when malaria was almost eliminated in Bhutan. It is a retrospective study.
Response: Thanks for suggesting to change the objective of the study. We have revised the objective as “to quantify the temporal, spatial and spatial-temporal patterns of both species of malaria at sub-district in Bhutan using retrospective surveillance data”.
- If it is, I will expect Figure 1 to show the distribution of malaria cases on a yearly basis from 2010 to 2019. Nine maps were supposed to be produced instead of limiting this figure to a vague title of “Map of Bhutan with malaria transmitting districts and sub-districts.”
Response: Our purpose of Figure 1 was to show the malaria transmitting sub-districts in relation to other districts of Bhutan and does not show the malaria cases on a yearly basis. Therefore, we have created an appendix with the yearly malaria cases. In addition, Figures 3 and 4 show yearly cluster of Plasmodium falciparum and P. vivax.
Line numbers 144-146: Yearly annual parasite incidence (API) of sub-districts (cases divided of sub-district divided by population per 1,000) has been shown in the Supplementary Figures 1 and 2.
- This way, it was going to help us see whether Kulldorff’s spatial and space-time scan statistics were able to predict the clustering of malaria. The results, especially maps, are presented it is impossible.
Response: Kulldorff’s spatial and space-time scan statistics can be used to identify/detect the clustering and not to predict the clustering of malaria. This is a commonly used methods as outlined in the following papers.
Soto-Calle, V., Rosas-Aguirre, A., Llanos-Cuentas, A., Abatih, E., DeDeken, R., Rodriguez, H., ... & Speybroeck, N. (2017). Spatio-temporal analysis of malaria incidence in the Peruvian Amazon Region between 2002 and 2013. Scientific reports, 7(1), 1-13.
Coleman, M., Coleman, M., Mabuza, A. M., Kok, G., Coetzee, M., & Durrheim, D. N. (2009). Using the SaTScan method to detect local malaria clusters for guiding malaria control programmes. Malaria journal, 8(1), 1-6.
- I agree with you that malaria is a vector-borne disease. Therefore, the life cycle of the mosquito depends on bio-climatic conditions. Its habitat varies in size. It may breed in numerous small pools of water that form due to rainfall. The larvae develop within a few days, escaping their aquatic environment before it dries out. As such, it is difficult, if not impossible, to predict when and where the breeding sites will form, and to find and treat them before the adults emerge.
If you agree with the above reasoning, why you did not take this into account? Your study has focused only on people who have tested positive. As you, also suggested people can migrate from one country to another and from one sub-district to another. Your clusters may not tell the entire story.
Response: We agree that vectors play role in the transmission. However, this study aimed to identify high malaria risk areas. Next step would be to identify the underlying reasons for higher risk of transmission.
- I have the impression that you relied on p-values to validate your results. Please read this link https://www.amstat.org/asa/files/pdfs/p-valuestatement.pdf to evaluate the statistical significance and p-values.
Response: Thanks for the information, it was an interesting article.

Reviewer 2 Report
- Research questions are not unclear. I can understand your argument that why spatiotemporal pattern of malaria incidence could help to provide policy implications about mosquito elimination, but people who are not working on GIS may not understand. Also, why is it important to see the seasonal pattern? Please highlight the importance of your analysis for disease outbreak context in the introduction.
- More comprehensive literature review is needed. There are a couple of studies using space-time analysis for malaria incidence. Through the literature review, the authors could address the challenges that have not been addressed yet.
Soto-Calle, V., Rosas-Aguirre, A., Llanos-Cuentas, A., Abatih, E., DeDeken, R., Rodriguez, H., ... & Speybroeck, N. (2017). Spatio-temporal analysis of malaria incidence in the Peruvian Amazon Region between 2002 and 2013. Scientific reports, 7(1), 1-13.
Coleman, M., Coleman, M., Mabuza, A. M., Kok, G., Coetzee, M., & Durrheim, D. N. (2009). Using the SaTScan method to detect local malaria clusters for guiding malaria control programmes. Malaria journal, 8(1), 1-6.
- Ethical considerations does not have to be included in Method section. It should be included in Institutional Review Board Statement. Please read this (https://www.mdpi.com/journal/ijerph/instructions)
Author Response
Reviewer #2
- Research questions are not unclear. I can understand your argument that why spatiotemporal pattern of malaria incidence could help to provide policy implications about mosquito elimination, but people who are not working on GIS may not understand. Also, why is it important to see the seasonal pattern? Please highlight the importance of your analysis for disease outbreak context in the introduction.
Response: The reasons for knowing the seasonality has been added in the revised manuscript.
Line numbers 64-66: In addition, knowledge of seasonal patterns will aid in the estimation of suitable period for transmission for initiating appropriate control measures (1, 2).
- More comprehensive literature review is needed. There are a couple of studies using space-time analysis for malaria incidence. Through the literature review, the authors could address the challenges that have not been addressed yet.
Response: The main limitations of study by Soto-Calle et al. (2017) was that analysis was done at a lower resolution at district level. The authors suggested the analysis to be done at higher resolution at a community level. In this study we have undertaken space-time analysis at a sub-district level. Our analysis is more realistic than other papers because the climatic conditions and other factors that support transmission within a sub-district are more likely similar as compared to district level. Therefore, our findings could be more accurate reflection of transmission risks in the area that can be useful to inform and devising of targeted control measures.
Line number 274-276: However, the main strength of our study is the breakdown of the monthly malaria incidence stratified by species at a fine-resolution (sub-districts), providing a more precise space-time analysis.
- Ethical considerations does not have to be included in Method section. It should be included in Institutional Review Board Statement. Please read this (https://www.mdpi.com/journal/ijerph/instructions)
Response: Ethical consideration has been deleted as suggested by the reviewers. Thanks you.
- Stuckey EM, Smith T, Chitnis N. Seasonally dependent relationships between indicators of malaria transmission and disease provided by mathematical model simulations. PLoS computational biology. 2014;10(9):e1003812.
- Nguyen M, Howes RE, Lucas TCD, Battle KE, Cameron E, Gibson HS, et al. Mapping malaria seasonality in Madagascar using health facility data. BMC medicine. 2020;18(1):26.

Round 2
Reviewer 1 Report
I agree with your answers to my comments and questions. However, you said that you created an appendix with the yearly malaria cases, unfortunately I did not see the appendix.
The structure of your manuscript follows the well established structure of manuscripts. The one you suggested (Spatio-temporal analysis of malaria incidence in the Peruvian Amazon Region between 2002 and 2013) did not. It has however, the merit of stating the null hypothesis and highlighted the difficulty of eradicating malaria.
Would you please clarify what you added to the paragraph (Line 186-188)
